# Centrosomal Protein 55 Regulates Chromosomal Instability in Cancer Cells by Controlling Microtubule Dynamics

**DOI:** 10.3390/cells13161382

**Published:** 2024-08-20

**Authors:** Stefanie Muhs, Themistoklis Paraschiakos, Paula Schäfer, Simon A. Joosse, Sabine Windhorst

**Affiliations:** 1Department of Biochemistry and Signal Transduction, University Medical Center Hamburg-Eppendorf, Martinistrasse 52, 20246 Hamburg, Germany; s.muhs@uke.de (S.M.); themistoklis.paraschiakos@gmail.com (T.P.); p.schaefer@uke.de (P.S.); 2Department of Tumor Biology, University Medical Center Hamburg-Eppendorf, 20246 Hamburg, Germany; s.joosse@uke.de; 3Mildred Scheel Cancer Career Center HaTriCS4, University Medical Center Hamburg-Eppendorf, 20246 Hamburg, Germany

**Keywords:** microtubule stability, CEP55, chromosomal instability, cancer, midbody

## Abstract

Centrosomal Protein 55 (CEP55) exhibits various oncogenic activities; it regulates the PI3K-Akt-pathway, midbody abscission, and chromosomal instability (CIN) in cancer cells. Here, we analyzed the mechanism of how CEP55 controls CIN in ovarian and breast cancer (OvCa) cells. Down-regulation of CEP55 reduced CIN in all cell lines analyzed, and CEP55 depletion decreased spindle microtubule (MT)-stability in OvCa cells. Moreover, recombinant CEP55 accelerated MT-polymerization and attenuated cold-induced MT-depolymerization. To analyze a potential relationship between CEP55-controlled CIN and its impact on MT-stability, we identified the CEP55 MT-binding peptides inside the CEP55 protein. Thereafter, a mutant with deficient MT-binding activity was re-expressed in CEP55-depleted OvCa cells and we could show that this mutant did not restore reduced CIN in CEP55-depleted cells. This finding strongly indicates that CEP55 regulates CIN by controlling MT dynamics.

## 1. Introduction

The thymus- and testis-specific scaffold protein Centrosomal Protein 55 (CEP55) is overexpressed in many types of cancer, including ovarian cancer, where it is associated with unfavorable prognosis [1,2,3,4]. Moreover, it has been widely demonstrated that this CEP55 overexpression increases the malignancy of cancer cells by promoting cellular abscission, proliferation, or chromosomal instability (CIN) (reviewed in [2]). These oncogenic activities are mediated by the scaffolding activity of CEP55. The protein can stimulate the PI3K-Akt pathway by directly interacting with the regulatory p85 subunit of PI3K [5,6,7,8], and in Hela cells, CEP55 is essential for the recruitment of Tumor Susceptibility Gene 101 (TSG101) and ALG-2-interacting protein X (Alix) to the midbody [9,10,11,12]. In turn, Alix recruits Endosomal Sorting Complexes Required for Transport III (ESCRT-III) proteins to the midbody and thereby initiates cellular abscission. Therefore, in Hela cells, CEP55 knock-down blocks midbody abscission [11]. However, in normal cells, CEP55 only controls midbody abscission in neuronal precursors, indicating that the effect of CEP55 on midbody abscission is cell-type specific [13,14].

In addition, CEP55 is essentially involved in the control of aneuploidy and CIN [15]. CEP55 belongs to the CIN70 score in cancer patients [16], and overexpression of CEP55 in mice increased spontaneous tumorigenesis as well as the CIN rate in mouse embryonic fibroblasts (MEFs) [8]. In CIN cells, chromosomes are not equally distributed to the daughter cells, resulting in aneuploid cell populations with highly diverse chromosome numbers. This genomic plasticity leads to selection of cells adapted to environmental stress, like chemotherapy or changes in the microenvironment during the process of metastasis [15]. Therefore, a high CIN score is associated with malignancy of tumor cells [16]. Mechanistically, CIN results from defects in the DNA repair machinery or from misregulation of microtubule (MT) spindle dynamics, leading to misaligned chromosomes in metaphase, and subsequently to chromosome missegregation in anaphase. These failures often result in the formation of lagging chromosomes or chromosome bridges [17,18], which after cellular division can be embedded by membranes and established as micronuclei [19,20].

In this study, we show that a high CEP55 level increases the CIN rate in OvCa and in breast cancer cells and our results from cellular and biochemical experiments indicate that CEP55 controls CIN by regulating MT stability.

## 2. Materials and Methods

### 2.1. Mammalian Cell Culture and Stable Lentiviral Knock-Down of CEP55

SKOV-3 cells were cultivated in McCoy’s 5A medium (26600, Gibco, Darmstadt, Germany) containing 10% (*v*/*v*) FCS, 2 mM Glutamine, and Pen/Strep (100 units/mL penicillin, 100 µg/mL streptomycin). SKOV-3 cells are derived from a human ascites metastasis of ovarian serous cystadenocarcinoma; mutation status: APC, FBXW7, PI3KCA, TP43. OVCAR-8 cells were cultivated in RPMI medium (72400, Gibco, Darmstadt, Germany) containing 10% (*v*/*v*) FCS and Pen/Strep (100 units/mL penicillin, 100 µg/mL streptomycin), OVCAR-8 cells are derived from human high-grade ovarian serous adenocarcinoma; mutation status: CTNNB1, ERBB2, KRAS, TP53. MDA-MB-231 cells were cultivated in DMEM medium (41965039, Gibco, Darmstadt, Germany) containing 10% (*v*/*v*) FCS and Pen/Strep. MDA-MB-231 cells are derived from a lung metastasis of breast adenocarcinoma; mutation status: CDKN2A, CDKN2B; BRAF, KRAS, TERT, TP53. MDA-MB-468 cells were cultivated in DMEM medium (41965039, Gibco, Darmstadt, Germany) containing 10% (*v*/*v*) FCS and Pen/Strep. MDA-MB-468 cells are derived from a lung metastasis of breast adenocarcinoma; mutation status: PTEN, RB1, TP53. Both MDA-MB-231 and MDA-MB-468 are triple negative (do not express estrogen and progesterone receptors and hormone epidermal growth factor receptors 2). All cell lines were purchased from the American Type Culture Collection (ATCC) and were authenticated by DSMZ (Deutsche Sammlung von Mikroorganismen und Zellkulturen GmbH, Leibniz-Institut, Braunschweig, Germany).

In all these carcinoma cell lines, a stable knock-down of CEP55 was performed as described before [21]. Vectors from the mission shRNA system from Sigma-Aldrich (Darmstadt, Germany) were used for lentiviral transduction. Five different CEP55 shRNA TRC2-pLKO.1 lentiviral vectors were tested. After selection with puromycin (3 µg/mL), the two cell lines with the strongest knock-down were selected (CEP55 sh1 and sh2). The targeted sequence of sh1 (TRCN000006197) is CCGGGCAGGCATGTACTTTAGACTTCTCGAGAAGTCTAAAGTACATGCCTGCTTTTTG. Sh2 (TRCN0000061975) targets the sequence CCGGGCAGCATCAATTGCTTGTAATCTCGAGATTACAAGCAATTGATGCTGCTTTTTG. Cells stably expressing scrambled shRNA were used as a control.

All used cell lines were tested on a regular basis for mycoplasma contamination, to ensure experiments with mycoplasma-free cells.

### 2.2. Stable Lentiviral Re-Expression of CEP55^WT^ and CEP55^59–428^

To generate the MT-binding, CEP55^59–428^ mutant quick-change mutagenesis was performed. For stable re-expression of CEP55^WT^ and CEP55^59–428^, the genes were cloned into the LeGo-iB_2_ Neo+ vector, and the empty vector was used as a control (a friendly gift from Kristoffer Riecken, UKE, Hamburg, Germany). Stable lentiviral re-expression of the proteins into CEP55 sh1 cells was performed as described [22]. Seventy-two hours after the last virus infection, the cells were selected using 700 µg/mL G418. Re-expression was analysed by Western blotting.

### 2.3. Western Blotting

Western blot analysis was performed by a standard procedure using nitrocellulose membranes. For CEP55 (anti-rabbit, #81693, Cell Signalling, Danvers, MA, USA) and GFP (anti-mouse, 11814460001, Roche, Basel, Switzerland) antibodies, the membranes were blocked for 30 min RT with 5% milk powder in Tris-buffered saline and Tween20 (TBS-T). For β-Tubulin (anti-mouse, T4026, Sigma, Darmstadt, Germany), 2.5% BSA/TBS-T was used for blocking. HSC70 (anti-mouse, sc-7298, Santa Cruz, Dallas, TX, USA) worked with both blocking methods. The secondary antibodies against mouse (ab205719, abcam) or rabbit (ab205718, Abcam, Cambridge, UK) were diluted in TBS-T and incubated for 1 h at RT. For signal production and detection, a chemiluminescence reagent (Amersham ECL Prime Western Blotting Detection Reagent, GE Healthcare Bio-Sciences, Amersham, UK) and the Intas ECL CHEMOCAM imager were used. Band intensities were quantified by Fiji (NIH National Institutes of Health) and normalized to HSC70.

### 2.4. Real Time PCR

To analyze the mRNA expression level of CEP55, a standard quantitative RT-PCR (qPCR) was performed. From cell pellets, the whole mRNA was extracted and cDNA generated using oligo dT primers. Samples were analyzed in triplicates with the QuantStudio™ 3 System (AppliedBiosystems, Waltham, MA, USA). CEP55 mRNA expression was evaluated using the following primers: CEP55 FW (GTCTGCTGCAACCTCACGAA) and CEP55 RV (AGCAGTTTGGAGCCACAGTC). GAPDH was used as internal control (GAPDH FW: AGTCCCTGCCACACTCAG, GAPDH RV: TACTTTATTGATGGTACATGACAAGG). The relative CEP55 expression was calculated using the ΔΔCt method and normalized to scr + L control cells.

### 2.5. Immunofluorescence Staining

An amount of 2 × 10^4^ cells seeded into chamber slides (Ibidi, Gräfelfing, Germany) were fixed in ice-cold methanol at −20 °C and incubated for 10 min. The washing steps were performed with 0.05% Triton-X-100/PBS. Cells were blocked with 1% BSA in 0.05% Triton-X-100/PBS for 1 h. The primary antibodies were diluted 1:200 in blocking solution and incubated for 1 h at room temperature. For co-staining, the β-tubulin antibody (T4026, Sigma) was used after the detyrosinated α-tubulin (ab48389, Abcam) or CEP55 antibody (ab48389, Abcam). Secondary antibodies were anti-rabbit Alexa-fluor^®^488 (A21206, Life Technologies, Darmstadt, Germany) and anti-mouse Alexa-fluor^®^568 (A11031, Life Technologies), both diluted 1:2000 in PBS and incubated for 1 h at RT. DNA was stained by 4′,6-diamidino-2-phenylindole (DAPI) in a 1:2000 dilution, 2 min RT. Images were obtained with Olympus IX83 (Olympus, Hamburg, Germany) or with Leica TCS SP8 X microscope (Leica, Wetzlar, Germany).

### 2.6. EB3 Tracking for Evaluation of MT Velocity

To assess MT velocity, cells were transfected with GFP-EB3 to label the plus ends of MTs in living cells. Cells were transfected with 75 ng of DNA, using the Biontex K2 transfection system (Biontex, Münschen, Germany), and incubated for 24 under standard conditions. Live cell imaging was performed using the Olympus IX83 microscope with an incubation chamber (37 °C, 5% CO_2_ and humidity). Z-stacked (5 × 0.2 µm) images were obtained every two seconds over a period of three minutes. Before MT velocity could be analyzed with TrackMate [23], the files were pre-processed using Fiji. In the first step, maximum intensity z-projection was performed. Next, the Difference of Gaussians filter plugin was used to remove background. In TrackMate, the DoG detector was employed to detect the EB3 comets. For particle tracking, the simple LAP tracker and filter for track duration and number of spots on track were used. The mean speed of the individual tracks was used to calculate the mean MT-velocity per cell. Thirty cells per group were examined. Statistical analysis was performed using the Mann–Whitney test.

### 2.7. Cold-Induced Depolymerisation in Cellulo

To assess the effect of CEP55 on MT stability in cells, MTs were depolymerized using cold treatment. Cells were seeded and cultivated as described above. For MT depolymerisation, cells were incubated with 4 °C cold culture medium for 30 min at 4 °C. Immediately afterwards, cells were fixed with cold 4% paraformaldehyde/sucrose for 10 min at room temperature and stained with antibodies as described above. Images were captured using an Olympus IX83 microscope.

### 2.8. Chromosome Isolation

Chromosomes were isolated as described [24]. Thereafter, chromosomes were analysed by microscopy and the number of chromosomes per nucleus was determined.

### 2.9. Next-Generation Sequencing

DNA isolated from the control and CEP55 knock-down SKOV-3 and OVCAR-8 cells underwent PCR-free, DNA nanoball, Next Generation Sequencing (BGI Genomics, Copenhagen, Denmark). Sequencing data were processed as before [25] with minor adjustments: fastq files were processed by fastp [26], followed by alignment to the hg38 reference genome using bwa-mem2. Copy number alterations (CNA) were estimated using control-FREEC’s segmentation algorithm [27]. Statistical analyses and visualization were performed using R version 4.1.1 (R Foundation for Statistical Computing) and In-Silico Online version 2.3.1 at http://in-silico.online (accessed on 23 August 2023).

### 2.10. Structure Prediction of CEP55

The secondary structure of the CEP55 protein was predicted by using different prediction programs. For evaluation of α-helix, β-sheet, and random coil probability, PSIPRED, PredictProteinOpen, NetSurfP-1.1, NetSurfP-2.0, and RaptorX SS8 were used, and for prediction of coiled-coil structures, DeepCoil1, DeepCoil2, Marcoil, and Ncoil [28,29,30,31,32,33,34,35,36]. The 3D structure of the CEP55 protein was predicted from its amino acid sequence using the DeepMind program AlphaFold [37].

### 2.11. Recombinant Expression of His-GFP-CEP55 and CEP55 Mutants

CEP55 gene was codon optimized for expression in *E. coli* strains and cloned into the psf421 vector to get His-GFP fusion proteins. Mutations were inserted by SLIC cloning.

Proteins were expressed in Rosetta 2(DE3)pLysS *E. coli*. The bacteria were incubated in Terrific Broth (TB) medium, shaken at 37 °C until OD600 of one and protein expression was induced with 0.1 mM IPTG. After incubation overnight at 16 °C, bacteria were harvested (3700× *g*, 10 min, 4 °C) and washed with PBS. For cell lysis, the pellet was resuspended in lysis-buffer (20 mM HEPES pH 7.5, 400 mM NaCl, 1 mM EDTA, 1 mM DTT, 0.5 mM benzamidine). After disruption with the cell homogenizer (Constant Systems Ltd., Daventry, UK) at 1.8 kbar, 1 mM PMSF and 0.1% Triton-X-100 were added. Prior to centrifugation (48,000× *g*, 30 min, 4 °C) 25 mM Imidazole was added and incubated for 5 min. The protein lysate was incubated with equilibrated Ni-NTA Agarose (R901, Invitrogen, Waltham, MA, USA) for 1 h at 4 °C. Beads with lysate were applied to a column and washed with 20 mM HEPES pH 7.5, 400 mM NaCl, 0.1% Triton-X-100, and 25 mM Imidazole. Followed by incubation with 10 mM ATP and 20 mM MgCl_2_ in 20 mM HEPES pH 7.5, 400 mM NaCl and 25 mM Imidazole for 25 min. Beads were washed with increasing Imidazole concentrations (35, 50, 62.5, and 75 mM Imidazole). Elution was performed twice with 20 mM HEPES pH 7.5, 400 mM NaCl and 130, respectively, 150 mM Imidazole, and 1 mM PMSF. Proteins were dialysed overnight at 4 °C to 20 mM HEPES pH 7.5, 250 mM NaCl, and 1 mM DTT. The protein concentration of elution fractions was quantified with a BSA standard by SDS-PAGE stained with Roti Blue quick (4829.1, Roth, Karlsruhe, Germany). Proteins were frozen in liquid nitrogen and stored at −80 °C.

### 2.12. MT-Binding

To analyse binding of MTs to recombinant expressed GFP, GFP-CEP55, and GFP-CEP55 mutants, the proteins were bound to equilibrated GFP-Trap^®^ Magnetic Agarose Beads GFP (ChromoTek, Planegg, Germany). In a volume of 1 mL, 355 nM of protein were incubated for 1 h at 4 °C with 15 µL beads solution in 20 mM HEPES pH 7.5, 250 mM NaCl, and 1 mM DTT. After incubation, beads were washed twice with buffer to remove unbound protein. To assess binding of MTs to CEP55-coupled agarose beads, tubulin prepared from porcine brain (for protocol, see https://hymanlab.org/uploads/methods/Prep-of-tubulin-from-Porcine-brain.pdf, accessed on 18 July 2024) was applied and polymerized as follows: 0.15 µM taxol was added to the tubulin solution (545 nM in 80 mM PIPES pH 6.8, 2 mM MgCl_2_ and 1 mM EGTA, 1 mM GTP) and incubated for 5 min at 37 °C. The taxol concentration was increased every 5 min to 1, 5, and finally to 20 µM. After polymerisation, MTs were cooled to 4 °C, pH was adapted to 7.5, and MTs were incubated with the immobilised GFP proteins for 1 h at 4 °C. After incubation, the beads were washed 6 times with 500 µL of 50 mM HEPES/pH 7.5, 1 mM MgCl_2_, 1 mM EGTA, 2.5 mM KCl, and 0.5 mM DTT, and binding of MTs to CEP55 was analysed by Western blotting, employing an antibody against β tubulin.

### 2.13. MT-Pulldown

To assess binding of CEP55 to MTs, a classical pulldown assay was performed, adapted from the Cytoskeleton Microtubule Binding Protein Spin-down Assay Kit (BK029) (Cytoskeleton, Inc., Denver, TX, USA). Here, MT-bound CEP55 co-precipitates with MTs.

MT-polymerisation was performed as described above in general tubulin buffer (80 mM PIPES pH 6.8, 2 mM MgCl_2_, and 1 mM EGTA) and the MT solution was diluted to an end concentration of 400 ng/µL. CEP55 proteins were applied in a concentration of 200 nM. The proteins were carefully placed on top of a taxol supplemented glycerol cushion (general tubulin buffer with 60% glycerol and 20 µM taxol), and after centrifugation at 100,000× *g* and 23 °C for 40 min, the uppermost 50 µL of the supernatant were transferred into the sample buffer. The remaining cushion buffer was discarded and the pellet resuspended in sample buffer. Supernatant and corresponding pellets were analysed by Western Blotting employing a GFP antibody. For evaluation, the percentage of protein bound to tubulin was determined.

### 2.14. MT-Polymerisation

To analyse MTs by fluorescence microscopy, rhodamine-labelled tubulin (#TL590M, Cytoskeleton, Inc., Denver, TX, USA) was prepared as described [22]. A concentration of 250 nM tubulin and 250 nM CEP55 proteins were mixed and incubated at 37 °C for 30 min. After incubation, 10 µL of the tubulin-protein solution were pipetted on a chamber slide, again incubated for 10 min, and coated with 30 µL Fluoromount G (SouthernBiotech, Birmingham, AL, USA). The sample was imaged by confocal microscopy (Leica TCS SP8 X microscope (Leica, Wetzlar, Germany).

Analysis of MT-polymerisation was performed by a turbidity assay as described [22]. For this experiment, 237 nM CEP55 proteins (diluted in general tubulin buffer) and 3 µM self-prepared tubulin (see above) were mixed with 1 mM GTP, general tubulin buffer (80 mM PIPES pH 6.8, 2 mM MgCl_2_, and 1 mM EGTA) and 10% DMSO. Polymerisation was measured in a Tecan Infinite 200 reader as absorption at 340 nm at 37 °C.

### 2.15. Cold-Induced In Vitro MT-Depolymerisation

To analyze the effect of CEP55 on MT stability, the cold-induced MT-depolymerization assay was performed. Rhodamine-labeled, taxol-stabilised MTs were prepared as described [22]. A concentration of 250 nM of CEP55 proteins and 250 nM rhodamine-labelled tubulin were mixed and incubated for 10 min at room temperature. For cold-induced depolymerization, the CEP55-tubulin mixture was incubated at 4 °C for 10 min in the presence of 4 mM CaCl_2_. A volume of 10 µL of the tubulin-protein solution was applied to a chamber slide, coated with 30 µL Fluoromount G (SouthernBiotech), and imaged with the Olympus IX83 microscope.

## 3. Results

### 3.1. Depletion of CEP55 Decreases CIN

In order to analyze the effect of CEP55 on CIN in OvCa cells, its mRNA was stably down-regulated via shRNA in two different OvCa cell lines with high CEP55 expression OVCAR-8 and SKOV-3. Here, shRNA1 (sh1) strongly (between 70% and 87%), and shRNA2 (sh2) moderately (40–60%) reduced the CEP55 level (Appendix A). In addition, the specificity of CEP55 depletion was controlled by stably re-expressing the protein in CEP55 sh1 cells (Appendix A). For validation, we also stably knocked-down CEP55 in the two triple-negative breast cancer cell lines MDA-MB-231 and MDA-MB-468. In MDA-MB-231 cells, shRNA1, and in MDA-MB-468 cells, shRNA2, had the strongest knock-down effect (Appendix A).

The impact of CEP55 on CIN was analyzed by different assays. As a first readout, the number of cells with properly aligned chromosomes [38] was compared between control and CEP55 shRNA SKOV-3 cells. To distinguish between misaligned chromosomes and pro-metaphase cells, only chromosomes arranged in the equatorial plate were considered (see also [8]). Misalignment was defined by chromosomes not arranged in the equatorial plate (for example, see arrow in Figure 1A). This evaluation revealed that strong down-regulation of CEP55 significantly increased the number of cells with aligned chromosomes, while moderate down-regulation had no significant effect (Figure 1B). Moreover, re-expression of CEP55 reversed decreased misalignment in CEP55 shRNA SKOV-3 and OVCAR-8 cells, validating the specificity of the CEP55 knock-down effect (Figure 1C).

Since misaligned chromosomes can lead to chromosome missegregation during anaphase [39,40], we next compared the frequency of defective anaphase cells between control and OVCAR-8 sh1 cells. These included lagging chromosomes as well as chromosome bridges in anaphase. We found that the number of cells with defective anaphase was 2.4-fold higher in control compared to sh1 cells (Figure 1D,E). Likewise, the number of micronuclei, a consequence of lagging chromosomes or chromosome bridges [19], was reduced by 50% in SKOV-3 and OVCAR-8 sh1 cells, and this effect was rescued by re-expressing CEP55 (Figure 1E,F).

To show whether in breast cancer cells, CEP55 also controls CIN, chromosome alignment was compared between control and CEP55 knock-down MDA-MB-231 and MDA-MB-468 cells. Interestingly, in these cell lines, the effect of CEP55 depletion was even more pronounced compared to OvCa cells. After strong as well as after mild knock-down of CEP55, the number of cells with aligned chromosomes increased between 40% and 55% (Appendix A). Moreover, the number of cells with micronuclei decreased by 55% on average in CEP55 knock-down compared to control cells (Appendix A). Thus, in MDA-MB-231 and MDA-MB-468 cells, a weak CEP55 knock-down also reduces the CIN rate.

Since chromosome missegregation can result in a higher diversity of chromosome number [41], chromosomes were counted in SKOV-3 and OVCAR-8 CEP55 control and manipulated cells. SKOV-3 cells are highly chromosomally instable, having an average of 107 chromosomes per nucleus while OVCAR-8 cells are less chromosomally instable with a mean number of 52 chromosomes per nucleus. Also, chromosome number is highly diverse in SKOV-3 cells, and less so in OVCAR-8 cells. After CEP55 knock-down, the diversity of chromosome number was significantly decreased in SKOV-3 sh1 but not in sh2 cells (Figure 2A,B), and chromosome number was again highly diverse in cells re-expressing CEP55^WT^ (Figure 2C). In addition, the chromosome number was slightly but significantly reduced in CEP55 knock-down SKOV-3 cells compared to control cells. A similar result was obtained from OVCAR-8 cells (Figure 2C). Thus, a high level of CEP55 seems to increase the number and the diversity of chromosomes.

To further validate this result, chromosome copy number was determined by whole genome, low-coverage, next-generation sequencing of OvCa control and knock-down cells (see also Appendix A). For evaluation, the median chromosome copy number of control cells was determined, and this value was defined as zero-baseline (bold black line in Figure 2D). To determine the effect of CEP55 knock-down, the chromosome copy number of shRNA cells (SKOV-3 and OVCAR-8) corresponding to the same genomic loci were subtracted (shRNA-control). In case the alterations of the copy number were higher in the knock-down cells, then it was defined as “further away from baseline”, while alterations higher in the control cells are described as (“closer to baseline”).

In SKOV-3 cells, a total of 66.9% (481.5 Mbp) of the differences between knock-down and control was closer to baseline after knock-down, whereas 33.1% (238.5 Mbp) was further away from baseline. Similarly, in OVCAR-8 cells, a total of 58.7% (229.9 Mbp) was closer to copy number baseline after knock-down, whereas 41.3% (162.0 Mbp) was further away from baseline. However, the chromosomes affected by CEP55 knock-down were different between SKOV-3 and OVCAR-8 cells; in SKOV-3 cells it was chromosomes 1, 4, 11, 15, 16, 20 and 22, and in OVCAR-8 cells 3, 10, 12, 13, 18, and 20. Thus, CEP55 does not affect distribution of specific chromosomes.

Taken together, these results indicate that the heterogeneity of chromosomal aberrations within the ovarian cancer cell lines was reduced after knock-down of CEP55.

### 3.2. CEP55 Depletion Reduces the Stability but Increases the Dynamics of MTs

Our results depicted in Figure 1, Figure 2 and Figure 3 reveal that a high CEP55 level increases the frequency of misaligned and missegregated chromosomes, the number of micronuclei as well as the heterogeneity of chromosome number, strongly indicating that a high CEP55 concentration promotes CIN and aneuploidy.

Since chromosomal misalignment can result from dysregulated spindle MT-dynamics [41], and CEP55 has been shown to directly bind to MTs in vitro [42], we analyzed whether CEP55 may be involved in the regulation of spindle MT dynamics. For this, we analyzed spindle MT-stability in M-phase SKOV-3 and OVCAR-8 control and CEP55-manipulated cells by staining the cells with antibodies against detyrosinated, thus stabilized MTs [43]. Figure 3A shows representative images of antibody-stained cells, and Figure 3B Western blot results of the CEP55-level in manipulated SKOV-3 and OVCAR-8 cells. For evaluation, the ratio of fluorescence intensity derived from detyrosinated tubulin relative to β-tubulin was assessed. We found that CEP55 knock-down reduced the fraction of stable spindle MTs compared to controls and revealed that CEP55 re-expression rescued this effect (Figure 3C). Moreover, co-staining of CEP55 and MTs in metaphase cells showed a CEP55 localization at the spindle pole, as well as on spindles near the centrosomes (Figure 3D).

To analyze whether CEP55 also regulates growth of MTs, the velocity of GFP-EB3-labled MTs of control and CEP55 sh1 cells was measured by live cell imaging (Figure 3E). Evaluation of videos revealed that CEP55 knock-down cells show a 1.5-fold increased MT-growth (µm/min) compared to control cells. Thus, CEP55-mediated MT stabilization seems to decrease MT dynamics.

Next, we assessed if the high stability of spindle MTs in high CEP55 expressing control cells may protect MTs from cold-induced depolymerization. For this purpose, the cells were incubated at 4 °C, and stained MTs were analyzed by fluorescence microscopy. Thereafter, the number of metaphase cells with intact, separated, and destroyed spindles were compared between control and CEP55 knock-down cells (for example, see Figure 4A, left panel, and [44]). This analysis revealed that knock-down of CEP55 expression reduced the fraction of cells with intact spindle MTs by about 60% compared to control cells (Figure 4A, right panel). Moreover, CEP55 was localised to the remaining spindle microtubules (Figure 4B). From this result, we conclude that CEP55 shields the spindle MTs from cold induced depolymerization. Together, these results show that a high level of CEP55 increases stability and decreases dynamics of MTs.

### 3.3. Identification of CEP55 Peptides Required for MT-Binding

Our data show that a high CEP55 level increases CIN as well as spindle MT stability in OvCa cells. In order to reveal whether both findings are related, a CEP55 mutant with deleted MT-binding peptides has to be re-expressed in CEP55 sh1 cells and chromosome alignment and micronucleus formation assessed again. However, the CEP55 peptides required for MT-binding had not been identified yet and therefore this issue was addressed in this study.

Since MT-binding domains are not conserved in amino acid sequence or tertiary structure [45], we predicted the domain architecture of CEP55 using bioinformatics. CEP55 is a coiled-coil protein and so far, only the EABR domain has been characterized on a structural level [9]. Analysis of α-helix, β-sheet, random coil, and coiled-coil probability (see Appendix A) revealed that the secondary CEP55 structure mainly consists of α-helices, and coiled-coil (CC) domains, while β-sheets were only found in the C-terminal region. In addition, the 3-D structure of CEP55 was predicted by AlphaFold [37] (Appendix A). Based on the coiled-coil predictions and the resolved EABR-Domain, we predicted CEP55 as a dimer. In this model, the EABR domains are connected and form well-defined coiled-coil domains. The N- and C-termini, however, are structurally less complex with loops and short α-helices or ß-sheets.

Based on these predictions, functional segments were defined as indicated in Figure 5A.

Among these, the N-terminus (aa 1–58; CEP55^59–464^), the C-terminus (aa 429–464; CEP55^1–428^) and the N- and C-termini together (CEP55^59–428^) were deleted and analyzed by a dynabeads assay, assessing binding of MTs to immobilized CEP55 (see Section 2). CEP55^59–464^ and CEP55^1–428^ showed about 20% reduced binding to MTs, in comparison to the binding capacity of CEP55^FL^, while binding of MTs to CEP55^59–428^ was reduced by 60% (Figure 6B). The strongly reduced binding of CEP55^59–428^, lacking the N- and the C-terminal domains, was confirmed by an MT-pull-down assay and by fluorescence microscopy, both assessing binding CEP55 to MTs (Figure 6C,D). Thus, the N- and the C-terminus are required for MT-binding.

In order to show whether the isolated, linker connected N- and C-termini may bind to MTs, binding of CEP55^Δ59–428^ (see Figure 6A, last row, and Appendix A) was analyzed by the MT-pull down assay and by fluorescence microscopy (Figure 6C,D). However, this CEP55 mutant did not or only very weakly bind to MTs. According to our AlphaFold prediction, folding of the isolated, N- and C-terminus is different from full-length (FL) protein (Appendix A first and last panel). In the dimer FL protein the N- and C-termini do not interact with each other, thus are freely accessible to MTs. Inside the isolated dimeric N- and C-termini, however, the coiled-coiled domains seem to interact, impeding the accessibility to MTs.

Together, our data strongly indicate that binding of CEP55 to MTs is mediated by the N- and the C-terminal peptides.

### 3.4. Re-Expression of a MT-Binding Deficient CEP55 Mutant Does Not Rescue Reduced CIN of CEP55 shRNA Cells

The identification of the CEP55 peptides required for MT-binding has now enabled us to analyze the impact of CEP55 MT-binding on CIN. For this purpose, the N/C-terminus of CEP55 was deleted (CEP55^59–428^) and cloned into the lentiviral Lego vector. OVCAR-8 and SKOV-3 CEP55 shRNA cells were infected with the virus coding for CEP55^59–428^ (Appendix A). First, we confirmed that the CEP55^59–428^ mutant does not bind to spindle MTs, by staining metaphase cells with antibodies against tubulin and CEP55 (Figure 7A).

Thereafter, the role of CEP55 MT-binding on the CIN rate was analyzed. For this, chromosomal alignment and micronuclei formation were compared between CEP55 control, sh1 knock-down and cells re-expressing CEP55^59–428^. The results of these experiments revealed that re-expression of CEP55^59–428^ did not rescue the reducing effect of CEP55 depletion on CIN (Figure 7B,C). These data indicate that binding of CEP55 to MTs is required for its promoting effect on CIN.

### 3.5. Impact of CEP55 on MT-Dynamics In Vitro

In cells, a high level of CEP55 increases spindle MT-stability but this does not allow to conclude that CEP55 directly controls this process. To assess this, the effect of bacterial expressed CEP55 on MT-stability was examined in vitro by analyzing the effect of recombinant CEP55 on cold-induced MT-depolymerization. For this purpose, rhodamine-labelled MTs were depolymerized in presence of 4 mM CaCl_2_ at 4 °C in absence and presence of CEP55. The result of this experiment revealed that CEP55 protected MTs from cold-induced depolymerization (Figure 8A,B).

To show if this effect alters MT-polymerization, in vitro MT-polymerization was analyzed in presence and absence of CEP55 using a turbidity assay (Figure 8C,D). As a negative control, the CEP55^59–428^ mutant was employed. Thereby, we found that CEP55 increased the rate of MT-polymerization 4-fold while the mutant did not affect MT-polymerization (Figure 8E). The slowly increasing slope in the presence of CEP55, combined with the data of Fabbro et al. 2005 [12], showing that CEP55 does not nucleate MTs in cells, indicates that increased MT-polymerization in presence of CEP55 does not result from MT-nucleation. Instead, it is most likely the result of attenuation of MT-depolymerization (MT-stabilization).

Together our in vitro data show that CEP55 protects MTs from cold-induced polymerization by 80%, and increased the rate of MT-polymerization 4-fold (Table 1), indicating that CEP55 protects MTs from depolymerization.

## 4. Discussion

CEP55 has been first described by Fabbro et al., in 2005 [12] and in parallel by Martinez-Martinez-Garay et al., 2006 [46]. From this time to present, more than 195 studies have been published, demonstrating the oncogenic activity of CEP55, which in normal cells is mainly expressed in thymus and testis [12,46]. In most publications a high CEP55 level has been shown to be associated with unfavorable prognosis of breast and ovarian cancer patients [1,3,4,47]. However, for ovarian cancer patients, Li et al. [48] revealed the opposite, and Kaplan Meier curves depicted in the Human expression atlas (TCGA database) did not reveal a prognostic value of CEP55 for ovarian and breast cancer patients (Appendix A). However, for FIGO III and V-staged breast cancer patients a high CEP55 level significantly (*p* = 0.014) correlated with low survival probability (Human expression atlas; TCGA database, Appendix A). Thus, it is important to carefully characterize the patient cohorts before predicting the prognostic value of CEP55. The studies we refereed to provide detailed information [1,4] or combined publicly available data [3,47].

Many efforts have been undertaken to analyze the mechanism of how CEP55 drives tumorigenesis. The main findings are that CEP55 interacts with the p85 regulatory subunit of PI3KCA and thereby stimulates the PI3K-Akt pathway and cellular proliferation in different tumor types [5,6,7,8]. Furthermore, CEP55-mediated recruitment of TSG101 and Alix to the midbody is a crucial step for midbody abscission and cytokinesis of Hela cells [10,11,12]. In addition, CEP55 is essentially involved in the control of CIN and aneuploidy. CEP55 belongs to the CIN70 score [16] and MEFs derived from CEP55 overexpressing mice show an increased CIN rate [8]. However, in contrast to CEP55-mediated regulation of PI3K-signalling and midbody abscission, the mechanism how CEP55 regulates CIN has not been elucidated yet, and therefore was the main focus of this study.

Our data show that a high level of CEP55 increased the CIN rate in OvCa and in breast cancer cells (see Figure 1 and Figure 2), and since CIN can be caused by dysregulation of spindle MT-dynamics [17,41] we speculated that CEP55 may be involved in the regulation of spindle MT-dynamics. Indeed, we found that a high level of CEP55 increased spindle MT-stability in OvCa cells and protected MTs against cold-induced depolymerization. In line with these data, the MT-growth rate was increased in CEP55 knock-down compared to control cells.

From these data we conclude that a high level of CEP55 stabilizes MTs and thereby reduces MT-dynamics (catastrophe and rescue [49]), which may explain the increased abundance of misaligned and missegregated chromosomes in high CEP55 expressing control cells. In order to reveal whether CEP55-induced MT-stabilization and its effect on CIN indeed are related we identified the yet unknown CEP55 MT-binding sites inside N-terminal (aa 1–58) as well as in C-terminal (aa 429–464) peptides. The basic unstructured N-terminus of CEP55 exhibits typical features of an MT-binding peptide [45], while the C-terminal peptide does not. However, a mutant with deleted N-terminus showed reduced but not deficient binding to MTs, showing the involvement of the C-terminus for MT binding. Since MT-binding domains can be very diverse, cryogenic electron microscopy-based data are necessary to finally reveal the nature of the MT-CEP55 interaction [45].

The identification of the MT-binding peptides enabled us to validate the requirement of CEP55-MT binding for its promoting effect on CIN. For this purpose, a mutant with deleted MT-binding peptides (CEP55^59–428^) was re-expressed in OvCa shRNA cells to show whether this manipulation restores the increased CIN rate of control cells. Analysis of chromosome alignment and micronuclei formation revealed that this was not the case, although re-expression of CEP55^WT^ fully rescued decreased CIN of shRNA cells. Moreover, we found that the CEP55^59–428^ mutant did not bind to the mitotic spindle, showing that localization of CEP55 to the spindle apparatus depends on its MT-interaction. In conclusion, these data strongly indicate that CEP55 controls CIN by directly binding to and stabilizing MTs. This conclusion is supported by experiments where expression of the MT-depolymerizing kinesin KIF2B reversed the increased CIN rate in CEP55 overexpressing MEFs [8].

The data presented in this study as well as previous findings allow some assumptions about the different roles of CEP55 during mitosis. Fabbro et al. [12] revealed that at the onset of mitosis two specific CEP55 serine-phosphorylations induce translocation of CEP55 from the centrosome to the mitotic spindle. Inside the centrosome localization of CEP55 is not dependent on its binding to MTs [46] but here CEP55 interacts with the γ-tubulin-anchoring Kendrin-CG complex [12]. However, after translocation to the mitotic spindle, binding of CEP55 to the spindle apparatus depends on its interaction with MTs while in telophase when CEP55 moves to the midbody its interaction partner again changes to the Central spindling complex [42]. Thus, re-location of CEP55 during mitosis depends on different transient protein-protein interactions. Interestingly, the sites for Kendrin-CG, MT-, and centralspindlin-binding as well as for mitosis-specific phosphorylation are located inside the C-terminus of CEP55 [12,42], indicating that this domain determines the dynamic CEP55 localization during mitosis. Thus, it would be interesting to analyze if masking of the C-terminal CEP55 domain inhibits the oncogenic activity of CEP55.

## 5. Conclusions

In conclusion, a high CEP55 level in OVCAR cells increases the CIN rate by over stabilizing spindle MTs leading to misattachment of spindle fibers to chromosomes and thereby to chromosome misalignment and segregation.

## Figures and Tables

**Figure 1 cells-13-01382-f001:**
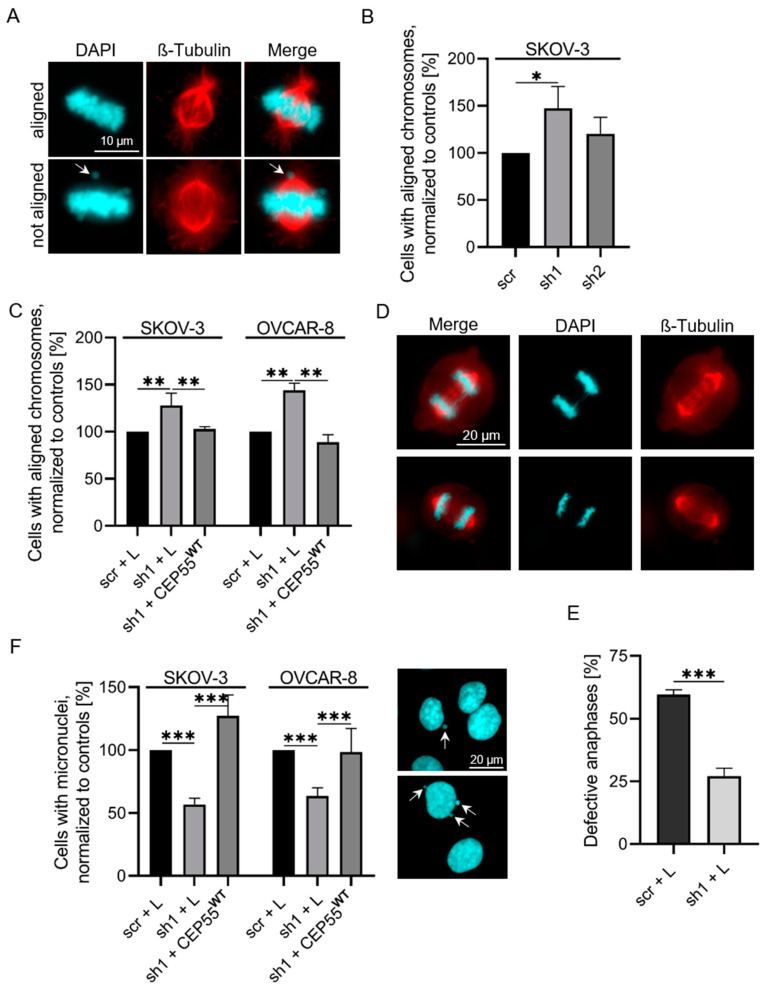
**CEP55 depletion decreases the CIN rate in OvCa cells.** (**A**) Metaphase cells were stained with Alexa-fluor568 coupled antibodies against β-tubulin (red) and with DAPI (blue). (**B**,**C**) The cells were grouped into cell populations with aligned or not aligned chromosomes and counted. The number of scrambled (scr) control cells with aligned chromosomes was set to 100%. (**D**,**E**) Fixed anaphase cells were stained with Alexa-fluor568 coupled antibodies against β-tubulin (red) and with DAPI (blue), and cells with normal and defective anaphases were determined. The graph shows the percentage of cells with defective anaphases in control (scr + L) and CEP55 knock-down (sh1 + L) cells. (**F**) Fixed interphase cells were stained with DAPI, the number of cells having micronuclei (indicated by arrows) were counted and the percentage of cells with micronuclei relative to the scrambled control samples (scr + L) was calculated. * *p* < 0.05, ** *p* < 0.001, *** *p* < 0.0001.

**Figure 2 cells-13-01382-f002:**
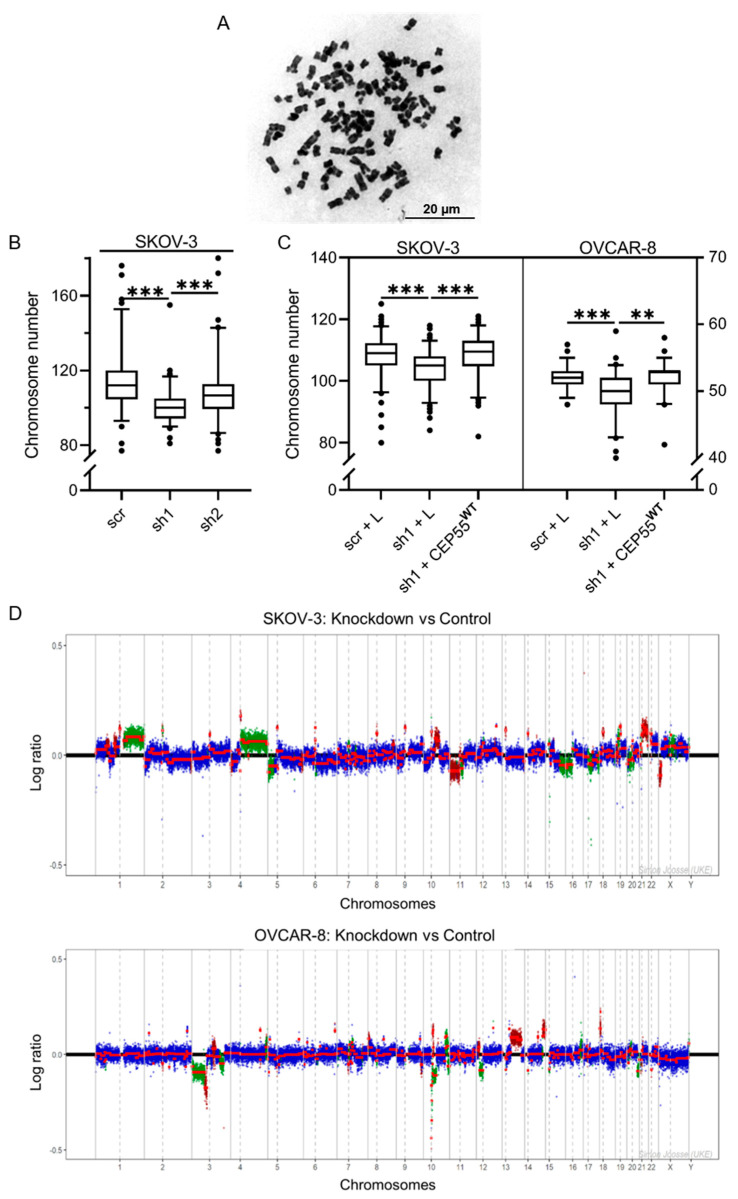
**Knock-down of CEP55 decreases genomic heterogeneity in OvCa cells.** (**A**–**C**) Chromosomes from metaphase control and CEP55 manipulated cells were prepared and chromosome number per cell was determined from at least 80 nuclei, obtained by three different preparations. Finally, chromosome heterogeneity was calculated by the F-test. ** *p* < 0.001, *** *p* < 0.0001. (**D**) In order to elucidate the differences in copy number alterations between CEP55 knock-down and control cells along the whole genome, log ratios corresponding to the same genomic loci were subtracted (knock-down minuscontrol). More extreme copy number alterations in the CEP55 knock-down cell lines as compared to the control cell lines (knock-down_gain_ > control_gain_|knock-down_loss_ < control_loss_), are described as “further away from baseline”, and are depicted as red shaded regions. However, if the copy number alterations were more extreme (i.e., further away from baseline) in the control as compared to the CEP55 shRNA cell, these regions were depicted green (knock-down_gain_ < control_gain_|knock-down_loss_ > control_loss_). For example: A chromosomal region with 5 copy numbers in the CEP55 knock-down and 4 copy numbers in the control cell lines, is depicted as a red positive value (5 − 4 = 1). A chromosomal region with 4 copy numbers in the CEP55 knock-down and 5 copy numbers in the control cell lines, is depicted as a green negative value (4 − 5 = −1). Green: regions closer to baseline in knock-down as compared to control. Red: regions further away from baseline in knock-down as compared to control. Blue: unchanged within 2 × ar(MedianRatio). The solid lines of the *X*-axis represent the chromosome borders, and the dashed line the centromeres.

**Figure 3 cells-13-01382-f003:**
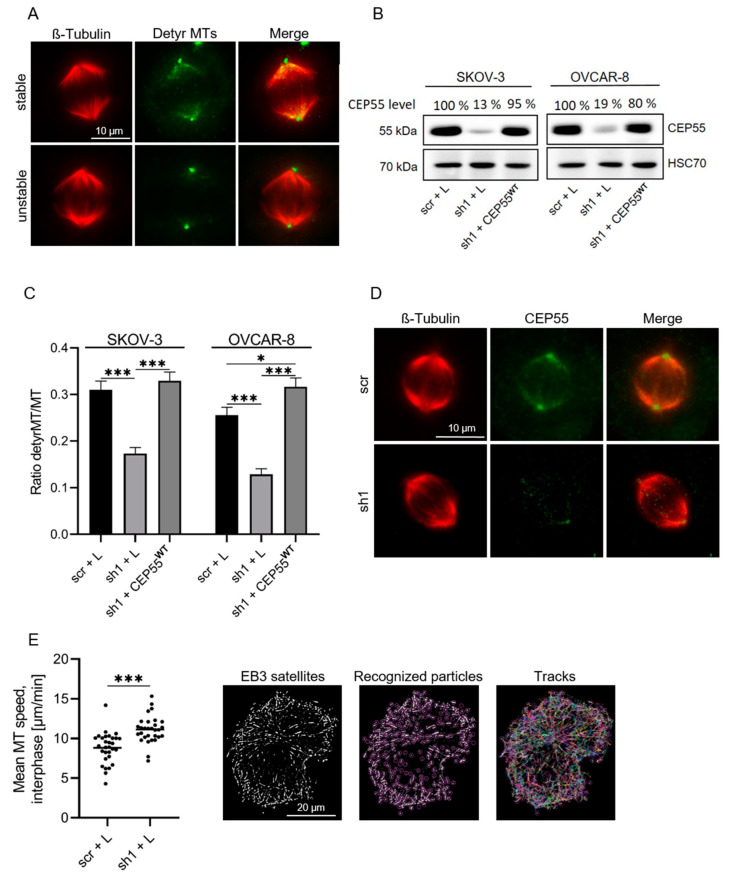
**Down-regulation of CEP55 in OvCa cells decreases stability of spindle MTs.** (**A**) Fixed SKOV-3 metaphase cells were stained with Alexa-Fluor568 coupled antibody against β-tubulin (red) and with Alexa-Fluor488 coupled antibody against detyrosinated tubulin (green). Bar: 10 µm. (**B**) The CEP55 level was analyzed in control (scr + L), CEP55 knock-down (sh1 + L) and CEP55 knock-down cells re-expressing CEP55 (sh1 + CEP55^WT^), band intensities were analyzed, normalized to the Hsc70 values, and the percentage relative to control (100%) calculated. (**C**) Fluorescence intensity derived from detyrosinated tubulin and β-tubulin was determined, and the ratio detyrosinated MTs to MTs was calculated. Displayed are mean values + SD of 30 cells of three different experiments. * *p* < 0.05, *** *p* < 0.0001. (**D**) Fixed SKOV-3 metaphase cells were stained with Alexa-Fluor568 coupled β-Tubulin antibody (red) and with Alexa-Fluor488 coupled antibody against CEP55 (green). Bar: 10 µm. (**E**) OVCAR-8 knock-down and control cells were transfected with EGFP-EB3, and the speed of EGFP-EB3 comets was analyzed from live cell imaging by using Fiji’s TrackMate plugin. Mean speed of the measured tracks per cell was calculated. Displayed is the mean MT speed (µm/min) of 30 cells per group *** *p* < 0.0001.

**Figure 4 cells-13-01382-f004:**
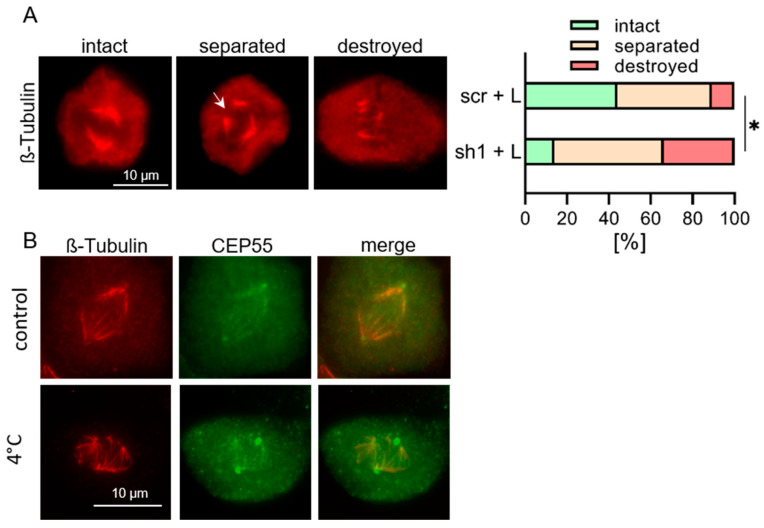
**Down-regulation of CEP55 in OvCa cells decreases stability of spindle MTs.** (**A**) OVCAR-8 metaphase cells were incubated at 4 °C for 30 min, fixed and stained with Alexa-Fluor568 coupled antibody against β-tubulin. Thereafter, the number of metaphase cells with intact, separated or destroyed spindle MTs (see image) were counted, and the percentage of each cell type determined. Separated MTs are marked by an arrow. Shown are mean values + SD from 20 cells across three different experiments. * *p* < 0.05. (**B**) OVCAR-8 metaphase cells were incubated at 4 °C for 30 min, fixed and stained with Alexa-Fluor568 coupled antibody against β-tubulin as well as with Alexa-Fluor488 coupled antibody against CEP55.

**Figure 5 cells-13-01382-f005:**
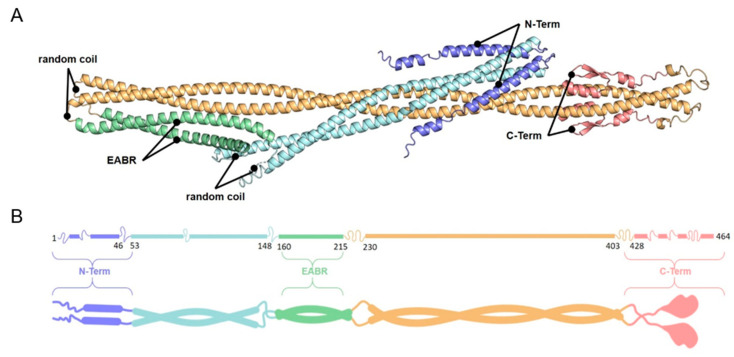
**Prediction of the CEP55 3-D structure.** (**A**) AlphaFold prediction of CEP55. (**B**) Cartoon showing the predicted CEP55 domains obtained by AlphaFold and bioinformatics (see Figure 5A and Appendix A).

**Figure 6 cells-13-01382-f006:**
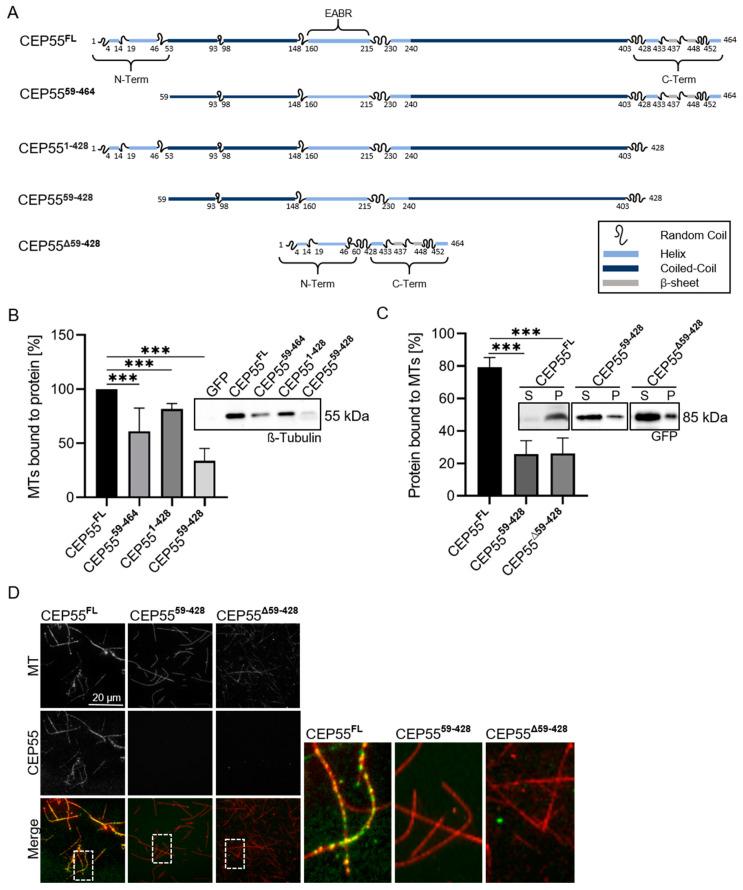
**The N- and C-terminus of CEP55 are essential for MT-binding.** (**A**) In order to identify the MT-binding domain inside the CEP55 molecule, CEP55 mutants were created as indicated in the Figure. (**B**) GFP, CEP55^FL^, or CEP55 mutants were coupled to GFP dynabeads and incubated with MTs. Binding of MTs to CEP55 was assessed by Western blotting. For evaluation MTs bound to CEP55^FL^ were set as 100%. The binding ability of CEP55 mutants was compared to the CEP55^FL^ reference. (**C**) MTs were pulled down by ultracentrifugation and the concentration of CEP55 in pellet (P) and supernatant (S) was analyzed by Western Blotting. The percentage of CEP55 in the pellet fraction was calculated. Shown are mean values + SD of three independent experiments. *p* < 0.005, *** *p* < 0.0001. (**D**) CEP55^FL^ or CEP55 mutants were co-incubated with rhodamine-labeled MTs (red) and analyzed by fluorescence microscopy. The right panel shows 5-fold magnification of the marked areas in the left panel.

**Figure 7 cells-13-01382-f007:**
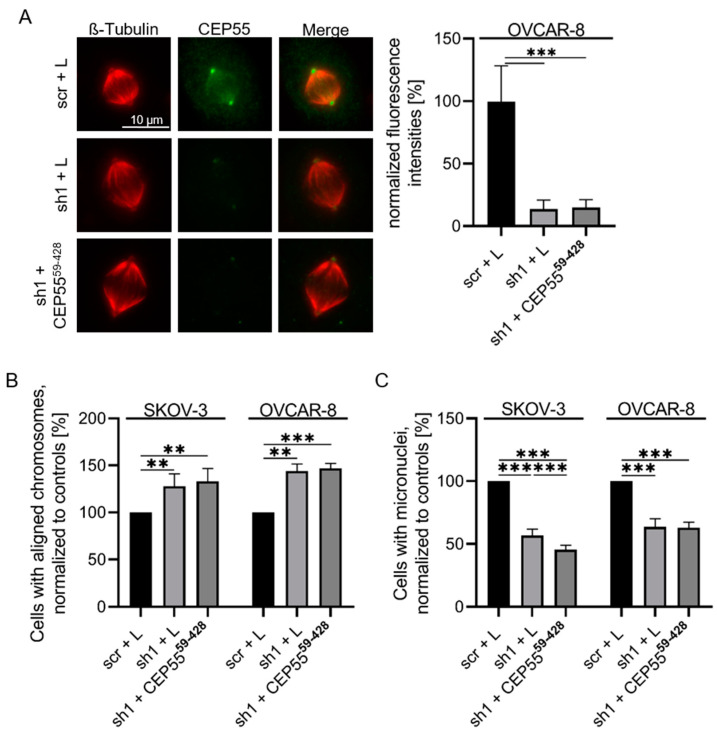
**Re-expression of a CEP55 mutant with deficient MT-binding activity does not restore decreased CIN of CEP55 knock-down cells.** (**A**) Fixed metaphase OvCa cells were stained with antibodies against CEP55 (green) and MTs (red), and spindle localization of CEP55 was analyzed by measuring fluorescence intensities derived from MT- and from CEP55-antibody. Finally, the ratio CEP55/MTs was calculated and the values obtained for control cells set to 100%. Shown are mean values + SD of at least 25 cells. *** *p* < 0.0001. (**B**) OvCa metaphase cells were stained with Alexa-fluor568 coupled antibodies against β-tubulin and DAPI. The cells were grouped in cell populations with aligned end or not aligned chromosomes and counted. (**C**) Percentage of micronuclei containing OvCa cells relative to the scrambled control samples (scr + L) was determined using DAPI stained interphase cells. ** *p* < 0.001, *** *p* < 0.0001.

**Figure 8 cells-13-01382-f008:**
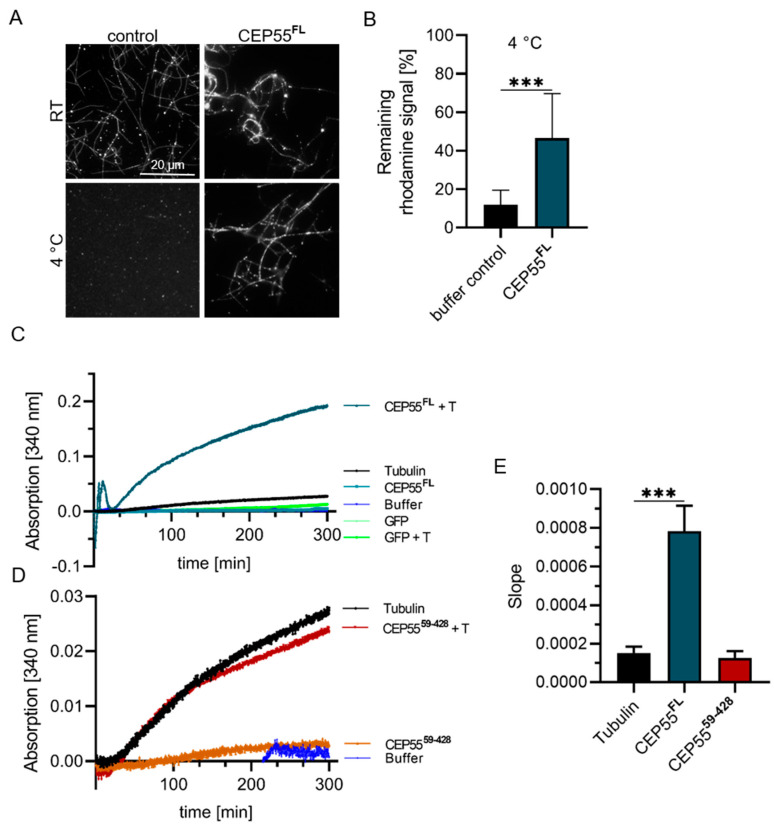
**CEP55 increases MT-polymerization in vitro.** (**A**) Rhodamine-labeled, taxol-stabilized MTs were incubated at RT or at 4 °C with 4 mM CalCl_2_ in presence of GFP or GFP-CEP55. (**B**) Evaluation of the remaining rhodamine fluorescence signal of cold treated samples. Fluorescence intensity was analyzed by Fiji using 8 different images, and remaining signal was normalized to RT control. (**C**,**D**) Polymerization of non-labeled, non-stabilized MTs was assessed by a turbidity assay in presence and absence of (**C**) CEP55 or (**D**) CEP55 MT-binding deficient mutant. In addition, all proteins in absence of MTs were measured as controls. (**E**) Mean values + SD of the calculated slope of three independent experiments are shown. *** *p* < 0.0001.

**Table 1 cells-13-01382-t001:** In vitro data on the effects of CEP55 on microtubule dynamics.

Method	Effect of CEP55	Extent
Cold-induced MT-depolymerization	protection	80%
MT-polymerization rate	slowly increasing slope	4-fold
conclusion	CEP55 attenuates MT depolymerization	

## Data Availability

The data that support the findings of this study are available in Figure 1, Figure 2, Figure 3, Figure 4, Figure 5, Figure 6, Figure 7 and Figure 8 and the Appendix A of this article.

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
