# Peer review of "Centrosomal Protein 55 Regulates Chromosomal Instability in Cancer Cells by Controlling Microtubule Dynamics"

_cells, 2024, doi:10.3390/cells13161382_

Round 1
Reviewer 1 Report
Comments and Suggestions for Authors
The manuscript of Muhs et al. investigates the role of CEP55 in regulating chromosomal instability (CIN) and microtubule dynamics using various cancer cell lines. Their main observations indicate that CEP55 downregulation leads to reduced spindle microtubule stability and CIN, and that a CEP55 mutant defective in microtubule binding, but not wt CEP55, fails to restore these effects. Based on these observations, the authors conclude that CEP55 influences microtubule stability, thereby causing CIN.
´Data from the Human Protein Atlas suggests that elevated CEP55 expression is associated with improved outcome in various cancer types. This seemingly contradicts the authors’ proposed role of CP55 in promoting CIN and cancer progression. Although the manuscript is well constructed, several key improvements are necessary. Addressing the following points will significantly strengthen the manuscript by providing a more comprehensive understanding of CEP55's role in cancer biology and its implications for CIN and microtubule dynamics.
1. Utilize information from the Human Protein Atlas to include Kaplan-Meier survival plots of CEP55 expression in various cancer types at the beginning of the results section. Add a discussion paragraph to reconcile these survival plots with the original results presented in the manuscript.
2. Perform immunoblot analysis to quantify the impact of CEP55 depletion on the levels of detyrosinated tubulin in OvaCa cells.
3. While the authors demonstrate that the N- and C-terminal segments of CEP55 are necessary for CEP55-mediated microtubule stabilization, it is crucial to investigate whether these segments are sufficient. Conduct experiments using GFP-fused termini to determine their sufficiency.
4. Include a table detailing in vitro data on the effects of CEP55 on microtubule dynamics, specifically focusing on rates of microtubule growth, shrinkage, and pausing periods.
Author Response
Comment 1. Utilize information from the Human Protein Atlas to include Kaplan-Meier survival plots of CEP55 expression in various cancer types at the beginning of the results section. Add a discussion paragraph to reconcile these survival plots with the original results presented in the manuscript.
Answer 1. We thank the reviewer for this suggestion. We referred to the data of Zhang et al., 2016; Kalimutho et al., 2018; Schiewek et. al., 2018, and Tang et al., 2020 clearly showing a correlation with high CEP55 and bad prognosis of ovarian or breast cancer patients. The fact that the TCGA data depicted in the Human Protein Atlas did not reveal significant correlations might be due to non-defined patient information. We discussed this issue on page 22.
Comment 2. Perform immunoblot analysis to quantify the impact of CEP55 depletion on the levels of detyrosinated tubulin in OvaCa cells.
Answer 2. We controlled our CEP55-manipulated cells every three weeks, and only worked with cells having a stable CEP55 knock-down or re-expression. Representative Western blots are now included in Figure 3B.
Comment 3. While the authors demonstrate that the N- and C-terminal segments of CEP55 are necessary for CEP55-mediated microtubule stabilization, it is crucial to investigate whether these segments are sufficient. Conduct experiments using GFP-fused termini to determine their sufficiency.
Answer 3: This is a very good point, we did this already but the results were not very clear. However, we agree that these results should be shown. Our data reveal that the isolated N- and C-terminus (connected by linker) did not bind to microtubule. The AlphaFold prediction (Figure S6, last panel) indicates that the flexibility of this peptide is too low to bind to microtubule. Please see Figure 6 C and D and page 17.
Comment 4. Include a table detailing in vitro data on the effects of CEP55 on microtubule dynamics, specifically focusing on rates of microtubule growth, shrinkage, and pausing periods.
Answer 4. Also this is a very good point, we did this. However, since we did not perform TIRF experiments (this method is not established in our group), we could not exactly evaluate growth, shrinkage, and pausing periods. In Table 1 our in vitro results shown in Figure 8 are summarized.

Reviewer 2 Report
Comments and Suggestions for Authors
In this manuscript entlited “Centrosomal Protein 55 regulates chromosomal instability in cancer cells by controlling microtubule dynamics” Muhs and collegues analyzed the mechanism of how CEP55 controls CIN in ovarian and breast cancer (OvCa) cells. CEP55 protein shows oncogenic activities and it is overexpressed in many types of cancer. It has been reported that CEP55 regulates the PI3K-Akt-pathway, cellular abscission and chromosomal instability (CIN) in cancer cells. The Authors performed several cellular and biochemical experiments showing that a high CEP55 level increases the CIN rate in OvCa and in breast cancer cells. Moreover, Muhs and Collegues results indicate that CEP55 controls CIN by regulating MT stability.
The Authors employed several different experimental approaches. The experimental work is well orchestrated, and the manuscript is well written. I really appreciate this paper, and due to the importance of CEP55 functions, I retain that it is surely of interest and worthy of being published.
I just have a minimal curiosity about Materials and Methods section.
2.5. Immunofluorescence staining page 3 lines 123-129.
In these sentences are described the fixation procedures for β-tubulin staining (4% paraformaldehyde/DMEM + 10% FCS) and for detyrosinated α-tubulin or CEP55 staining (ice-cold methanol at -20 °C). Based on this reviewer's experience ice-cold methanol at -20 °C is the most suitable procedure for microtubule fixation. Could the Authors clarify why they employed 4% paraformaldehyde/DMEM + 10% FCS to fix cells for β -tubulin staining?
Author Response
Comment 1 In these sentences are described the fixation procedures for β-tubulin staining (4% paraformaldehyde/DMEM + 10% FCS) and for detyrosinated α-tubulin or CEP55 staining (ice-cold methanol at -20 °C). Based on this reviewer's experience ice-cold methanol at -20 °C is the most suitable procedure for microtubule fixation. Could the Authors clarify why they employed 4% paraformaldehyde/DMEM + 10% FCS to fix cells for β -tubulin staining?
Answer 1 We thank the reviewer for noticing. We did not correctly describe this. Since we did co-stainings with β-tubulin and detyrosinated α-tubulin it was not possible to use two difference fixation procedures. All samples were fixed with ice-cold methanol. We corrected this mistake, please see page 5.

Round 2
Reviewer 1 Report
Comments and Suggestions for Authors
The auth unors have adressed adequately all my concerns.